

# Nutritional value of ensiled *Guizotia abyssinica* (L. f.) Cass. (Noug: Niger) treated with salt, molasses, urea or barley

Ayşe Gül Filik

Agricultural Biotechnology Department/Agriculture Faculty, Kırşehir Ahi Evran University, Kırşehir, Türkey

## ABSTRACT

The research aimed to assess the nutritional value of ensiled *Guizotia abyssinica* (L. f.) Cass. (GA), also known as Noug or Niger, when treated with various additives, including urea, salt, molasses, and barley. The eight treatment groups were as follows: (I) GA forage without any supplementation (control), GA with supplement (II) 1% salt, (III) 2.5% molasses, (IV) 2.5% urea, (V) 5% barley, (VI) 2.5% molasses + 1% salt, (VII) 2.5% urea + 1% salt, (VIII) 5% barley + 1% salt supplement with GA. Silage samples were analyzed using physical, chemical, and microbiological methods to determine their nutritional composition, sensory characteristics, and microbial content. The objective of this analysis was to provide a comprehensive and objective evaluation of the silage samples. Based on the assessment of relative feed value (RFV) and relative forage quality (RFQ), it was determined that GA forage supplemented with 2.5% urea produced the highest-quality silage. The findings suggest that GA forage has the potential to serve as a high-quality silage. It is recommended that GA forage be ensiled with various additives at different inclusion rates. Further research is required to establish its *in vitro* digestibility and assess animal preference *in vivo*.

# INTRODUCTION

Silage is the primary and most frequently used feed source in ruminant nutrition. When discussing silage, maize crops are typically the first to come to mind. However, growing maize for silage requires a water supply of 400–750 mm during the growing season (*Kuşvuran et al., 2015*). The global climate crisis has exacerbated drought conditions, increased water demand and prompting critical policy decisions regarding agriculture. Addressing these challenges requires expanding the cultivation of drought-tolerant crops that maintain high yield potential and nutritional value while requiring minimal water input. Additionally, rapid population growth and urbanization have led to competition for land resources between human food crops and animal feed crops. To mitigate this issue, researchers are increasingly focusing on crops that can thrive on marginal, arid, or poorly irrigated land, as opposed to traditional cereals and forages that are typically grown

Corresponding author
Ayşe Gül Filik,
aysegulcivaner@ahievran.edu.tr

on fertile agricultural land (*Degu, Melaku & Berhane, 2009*; *Filik & Filik, 2021a*). In recent years, the Food and Agriculture Organization (FAO) has incorporated traditional crops into its drought management agenda, emphasizing those capable of fulfilling the nutritional needs of both humans and animals in arid regions (*Filik, 2020*). The pseudo-cereal group, including amaranth, chia, finger millet, common buckwheat, quinoa, and teff, has been identified as a promising alternative to conventional crops (*FAO, 2019*). Beyond these crops, *Guizotia abyssinica* (L. f.) Cass. (GA), commonly known as Niger or Noug, has gained significant attention beyond its native Ethiopia and India in recent years. It is particularly well-suited for poorly ventilated and nutrient-deficient soils (*Alemaw & Wold, 1991*; *Alemayehu & Ashagrie, 1991*). Although some sources suggest its domestication around 3000 BC (*Hiremath & Murthy, 1988*), archaeobotanical evidence indicates its first recorded use between 800 BC and 700 AD during the Aksumite period (*Boardman, 1999*; *Boardman, 2000*).

GA is an annual dicotyledonous species with excellent biomass potential, reaching up to two m in height and yielding 340 kg of seeds per hectare. Its seeds contain approximately 40% oil, making it a valuable resource for biofuel production (*Getinet & Sharma, 1996*; *Gordin, Scalon & Masetto, 2015*). In Africa, GA is primarily cultivated in Ethiopia, with smaller-scale production in Sudan, Uganda, Tanzania, Malawi, and Zimbabwe (*Kurenkova, Tolkacheva & Zapivalov, 2024*). Unfortunately, GA remains largely unknown and underutilized in Türkiye, except for experimental studies (*Bahadır Koca et al., 2019*). The oil extracted from GA seeds has a light-yellow hue, a pleasant aroma, and a hazelnut-like flavor. Chemical analysis has shown its fatty acid composition to be approximately 75.4% linoleic acid, 9.7% palmitic acid, 6.9% stearic acid, 7.0% oleic acid, and less than 0.1% linolenic acid (*Alemaw & Wold, 1991*; *Alemaw & Wold, 1995*). *Makuria et al. (2025)* reported that linoleic acid was the predominant unsaturated fatty acid, ranging from 67.30% to 74.67% (179–234 mg/g), followed by oleic acid at 5.43% to 11.02% (1.03–1.60 mg/g of dry matter). Among saturated fatty acids, palmitic acid was the most abundant, comprising 10.32% to 10.66% (24.80–37.10 mg/g). Additionally, total phenolic content (TPC) varied between 10.89 and 11.78 mg GAE/g, while total flavonoid content (TFC) ranged from 5.42 to 6.67 mg CE/g, with aqueous methanol (80%) yielding higher phenolic content than absolute methanol. The DPPH scavenging assay IC50 values were recorded between 133 and 188 µg/mL, though they exhibited a weak correlation with TPC. GA seeds are consumed by humans due to their nutritional benefits, while the oil cake, left after oil extraction, is a valuable protein source for animal feed or biomass (*Kurenkova, Tolkacheva & Zapivalov, 2024*). The nutrient composition of GA seed meal was determined by *Yalew, Urge & Tadese (2024)* as follows: dry matter (DM) 93.01%, crude protein (CP) 34.85%, crude fiber (CF) 12.27%, ether extract (EE) 8.05%, crude ash (CA) 9.21%, and metabolizable energy (ME) 2,924.8 kcal/kg, respectively. It is widely used in the diets of honey bee, fish, monogastric animals (such as laying hens, broilers, and pigs), and ruminants (including calves, cattle, oxen, goats, sheep, and camels) (*Merrea et al., 2004*; *Duguma et al., 2004*; *Dekebo et al., 2004*; *Dessalegn et al., 2004*; *Galmessa et al., 2004*; *Bekele, Gojam & Shiferaw, 2004*; *Asfaw et al., 2004*; *Alemu, Melaku & Tolera, 2010*; *Asmare, Melaku & Peters, 2010*; *Kitaw, Melaku & Seifu, 2010*; *Alem, Tamir & Kurtu, 2011*;

*Tadesse et al., 2012*; *Nurfeta, Churfo & Abebe, 2013*; *Kebede & Tadesse, 2014*; *Chibsa et al., 2014*; *Abraham, Urge & Animut, 2015*; *Diba et al., 2015*; *Melesse et al., 2015*; *Abebe et al., 2017*; *Ayenew et al., 2019*; *Abreha et al., 2019*; *Adugna, Mekuriaw & Asmare, 2020*; *Bahadır Koca et al., 2019*; *Ali, Amha & Urge, 2020*; *Mengistu, Assefa & Tilahun, 2020*; *Yalew, Urge & Tadese, 2024*; *Kumsa et al., 2024*). Ruminant feed can comprise up to 49.5% GA seed meal (*Duguma et al., 2004*; *Dessalegn et al., 2004*; *Merrea et al., 2004*; *Mekonnen et al., 2019*). GA is cultivated for both seed and oil production, and its oil cake is becoming an increasingly important crude protein source for animal nutrition. Despite its potential, GA remains largely underutilized in scientific research, particularly concerning green herbage yield and silage production (*Getinet & Sharma, 1996*). Consequently, the findings of this study have the potential to be groundbreaking, attracting interest from numerous researchers and offering valuable new insights into GA's agricultural and nutritional potential.

## MATERIALS AND METHODS

### Silage material

The study was carried out in the agricultural experimental field area at Kırşehir Ahi Evran University in Türkiye. The research site is located at a longitude of 39°10′N and 34°22′E and at an altitude of 988 m above sea level. The cultivation of the GA plant occurred between May and September 2016. The monthly precipitation and temperature averaged from 0 to 98 mm and 8.1 to 34.68 °C, respectively. The seeds of *Guizotia abyssinica* (L. f.) Cass. (also known as Noug or Niger) were acquired from the Bahri Dağdaş International Agricultural Research Institute, the Ministry of Agriculture and Forestry in the Republic of Türkiye. The method of *Peiretti, Gai & Tassone (2015)* was used to cultivate the seeds and evaluate the nutrient levels of the GA plant at various stages.

### Silage preparation and fermentation

During the early flowering period, the GA plants were chopped using knives and cut into appropriate sizes for producing silage. The chopped plants were mixed uniformly with additives such as urea, salt, molasses, and barley. Subsequently, the mixtures were weighed and replicated four times. In totality, 32 silage samples were prepared and stored in bags measuring 200 × 250 mm with an oxygen permeability rate of 1.13 cc/m$^2$ day. Vacuum sealing of the samples was done using a laboratory-grade Packtech PT-VKM-CPRO machine. Treatments were prepared, consisting of the Control treatment and treatments treated with various combinations of GA, molasses, salt, urea, and barley as follows: 1% salt, 2.5% molasses, 2.5% urea, 5% barley, 2.5% molasses and 1% salt, 2.5% urea and 1% salt, and 1% salt and 5% barley. Silages that had been prepared were placed under scrutiny at the Enzyme and Microbial Biotechnology Laboratory, located within the Department of Agricultural Biotechnology at the Faculty of Agriculture in Kırşehir Ahi Evran University.

### Silage physical analyses

The temperature was maintained at 20 ± 2 °C. On the 90th day, the temperature was measured using a LYK 9263 digital thermometer, pH was measured with an Eutech Instruments pH 700, water-soluble carbohydrates were measured using a Hanna

Instruments HI 96801 Digital Refractometer with Brix degree ranging from 0 to 25°, and color was assessed *via* a Konica-Minolta CR-410 color meter, as referenced by *Magalhães et al. (2012a)*, *Magalhães et al. (2012b)*, *Filik et al. (2018)*, *Çayıroğlu et al. (2020)*, and *Filik & Filik (2021a)*. The Fleig Score (FS) was also measured. The quality of silages was assessed by calculating the dry matter (DM) and pH values in accordance with the formula outlined by *Kılıç (1986)*.

$$FS = 220 + (2 \times DM\% - 15) - 40 \times pH. \tag{1}$$

Silages scoring between 100 and 80 on the Fleig scale are regarded as excellent, while those between 80 and 61 are considered good, and those ranging from 60 to 41 are considered average in quality. Silage scoring between 40 and 21 is deemed of low quality, and silage with scores ranging from 20 to 0 is considered very poor (*Kılıç, 1986*). After conducting physical property assessments, the silage samples underwent drying in a ventilated BINDER ED115 oven at 65 °C for 48 h. Upon completion of the drying process, analysis for air dry matter was carried out. The dried samples were then ground, passing through a one mm sieve using the Ultra-Centrifugal Mill ZM 200 (Retsch, Haan, Germany) (*Filik & Filik, 2021b*).

## Silage chemical analyses and energy calculation

DM was determined using the 925.40 method, while organic matter (OM) was measured using the 934.01 method. CP was measured with the 984.13 method, ether extract (EE) with the 920.39 method, and CA was determined using the ash method 942.05 (4.1.10). AOAC procedures (*2006*) were employed for all determinations. CF, acid detergent fiber (ADF), neutral detergent fiber (NDF), and acid detergent lignin (ADL) were analyzed using the Ankom200 Fiber Analyzer by Ankom Technology Corp., located in Macedon, NY, USA, in accordance with the procedures outlined by *Van Soest, Robertson & Lewis (1991)*, *Ankom Technology (2016)*, *Ankom Technology (2017a)*, *Ankom Technology (2017b)*, and *Ankom Technology (2017c)*. Total carbohydrate (TC) content, non-fiber carbohydrate (NFC), and nitrogen-free extract (NFE) content (g/kg) were analyzed following AOAC procedures (*2006*). The procedures used were TC method BFM156 and NFC method BFM121. Total digestible nutrient (TDN%) values, digestible energy (DE Mcal/kg), metabolizable energy (ME Mcal/kg), net energy-lactation (NEL Mcal/kg), net energy-maintenance (NEm Mcal/kg) and net energy-gain (NEg Mcal/kg) values were calculated using silage nutrient analysis results (*Moe, Flatt & Tyrell, 1972*; *Heeney, 1978*; *Garrett, 1980*; *NRC, 2001*; *Schroeder, 2004*).

## Silage quality calculation and microbiological analyses

Dry matter intake (DMI), digestible dry matter (DDM), relative feed value (RFV), and relative forage quality (RFQ) were calculated using acid detergent fiber (ADF), neutral detergent fiber (NDF), and total digestible nutrients (TDN) data (*Rohweder, Barnes & Jorgensen, 1978*; *Linn & Martin, 1991*; *Undersander & Moore, 2002*; *Undersander, 2003*; *Kiliç & Abdiwali, 2016*; *Jeranyama & Garcia, 2004*). Total live bacteria (TLBc) and lactic acid bacteria count (LABc) were determined according to the methodology described

by *Harrigan (1998)*. Microorganism counts were done by plate counting, following the procedure outlined by *Cai et al. (1999)*. The meaning of the chemical quality values was determined by using four separate data points. Crude protein values and metabolizable energy were calculated from the chemical analysis results corrected on a dry matter basis. The RFV and RFQ were also determined based on the chemical analysis results.

## Statistical analysis

Descriptive statistics were analyzed using descriptive variables for the statistical analysis of the data. Descriptive statistics were calculated for eight different treatments, affecting quality, color, nutritional content, energy values, relative feed value and relative forage quality of GA silages. The normality of the data distribution was assessed using the Shapiro–Wilk's test, while variance homogeneity was evaluated with Levene's test. Statistical significance of mean differences between treatments was determined at ($p < 0.05$) or ($p < 0.01$). One-way ANOVA was used to test for differences between the means of GA silages (*Sheskin, 2004*; *Montgomery, 2008*). The SAS (*SAS, 2001*) statistical software package was used to compute the Standard Error of Mean (SEM), as well as the Tukey's HSD multiple range test (*Genç & Soysal, 2018*). To test linear, quadratic, and cubic effects resulting from GA silage treatments, single-degree-of-freedom polynomial contrasts were also employed (*Steel & Torrie, 1980*).

## RESULTS

### Quality and color characteristics of GA silages

The results of the study indicate that different additives had significant effects on the quality and color parameters of *Guizotia abbysinica* (L. f.) Cass. (GA) silages (Table 1). The silage temperature was significantly lower in groups containing 2.5% molasses (19.40 °C), 5% barley (19.50 °C), and 2.5% molasses + 1% salt (19.63 °C) compared to the control (22.95 °C) ($p < 0.0001$). The most important parameter during silage fermentation is the pH of the silage. In the current study, as given in Table 1, pH levels of silages decreased significantly with the addition of 2.5% molasses (4.20), 2.5% urea (4.21), and 2.5% molasses + 1% salt (4.13), indicating improved fermentation quality ($p < 0.0001$). The Fleig Score (FS), which reflects silage fermentation quality, was highest in silages containing 2.5% molasses + 1% salt (93.60) and 2.5% urea + 1% salt (93.61), showing a significant improvement compared to the control (69.82) ($p < 0.0001$). Regarding color parameters, the total color difference ($\Delta E^*$) was lower in silages containing 2.5% molasses + 1% salt (23.82) compared to the control (26.89). The lightness (L*) values were slightly decreased in the molasses and salt treatments, but the differences were not statistically significant ($p > 0.05$). Another parameter that is important in silage quality is water soluble carbohydrate (WSC), which was determined in the study as 15.50, 15.25, 15.75, 16.75, 16.00, 18.25, 18.50, and 15.50, respectively ($p < 0.05$).

### Nutrient composition of GA silages

The nutrient composition analysis (Table 2) demonstrated that dry matter (DM) content was significantly increased with the addition of 2.5% molasses + 1% salt (268.01 g/kg) and

Filik (2025), *PeerJ*, DOI 10.7717/peerj.19267

**Table 1  Quality and color of GA silages with different additives.**

| Parameters[1] | °C ± SEM | pH ± SEM | WSC (ºBrix) ± SEM | Fleig Score ± SEM | L* ± SEM | a* ± SEM | b* ± SEM | ΔE* ± SEM | h ± SEM | C ± SEM |
|---|---|---|---|---|---|---|---|---|---|---|
| Control (*Guizotia abbysinica* (L. f.) Cass. (GA)) | 22.95 ± 0.06A | 4.39 ± 0.02A | 15.50 ± 0.65c | 69.82 ± 0.40D | 25.36 ± 0.57 | 3.07 ± 0.36 | 8.34 ± 0.34a | 26.89 ± 0.62 | 69.79 ± 2.49 | 73.02 ± 5.74a |
| GA with 1% salt | 22.98 ± 0.05A | 4.38 ± 0.01A | 15.25 ± 0.48c | 73.24 ± 0.16D | 24.38 ± 0.26 | 2.90 ± 0.12 | 7.30 ± 0.12abc | 25.62 ± 0.25 | 68.32 ± 0.88 | 56.16 ± 1.70abc |
| GA with 2.5% molasses | 19.40 ± 0.04B | 4.20 ± 0.01B | 15.75 ± 0.85bc | 86.99 ± 0.95AB | 24.48 ± 0.56 | 3.39 ± 0.39 | 6.99 ± 0.18bc | 25.69 ± 0.63 | 64.39 ± 1.92 | 52.38 ± 2.86bc |
| GA with 2.5% urea | 22.90 ± 0.12A | 4.21 ± 0.04B | 16.75 ± 1.25abc | 82.79 ± 0.65BC | 25.56 ± 0.89 | 2.84 ± 0.28 | 7.97 ± 0.15ab | 26.92 ± 0.89 | 70.50 ± 1.60 | 66.35 ± 2.59ab |
| GA with 5% barley | 19.50 ± 0.09B | 4.40 ± 0.01A | 16.00 ± 0.71abc | 75.53 ± 3.24CD | 24.82 ± 0.47 | 2.86 ± 0.23 | 6.78 ± 0.25bc | 25.89 ± 0.52 | 67.18 ± 1.26 | 49.06 ± 3.66bc |
| GA with 2.5% molasses + 1% salt | 19.63 ± 0.12B | 4.13 ± 0.02BC | 18.25 ± 0.75ab | 93.60 ± 0.09A | 22.70 ± 0.11 | 2.74 ± 0.17 | 6.63 ± 0.46c | 23.82 ± 0.23 | 67.32 ± 1.93 | 47.27 ± 6.34c |
| GA with 2.5% urea + 1% salt | 23.10 ± 0.18A | 4.06 ± 0.09C | 18.50 ± 0.29a | 93.61 ± 2.91A | 24.38 ± 0.27 | 3.25 ± 0.23 | 7.68 ± 0.31abc | 25.78 ± 0.37 | 67.09 ± 1.04 | 62.53 ± 5.14abc |
| GA with 5% barley + 1% salt | 19.55 ± 0.10B | 4.18 ± 0.04B | 15.50 ± 1.04c | 86.41 ± 1.15AB | 25.34 ± 1.11 | 2.65 ± 0.28 | 7.48 ± 0.70abc | 26.57 ± 1.26 | 70.31 ± 2.12 | 60.05 ± 11.05abc |
| | *P* value | 0.0001 | 0.0001 | 0.0419 | 0.0001 | 0.0733 | 0.5272 | 0.0357 | 0.0767 | 0.2466 | 0.0457 |
| | L | 0.0001 | 0.0006 | 0.2486 | 0.0001 | 0.8092 | 0.8685 | 0.3870 | 0.9519 | 0.8185 | 0.3536 |
| Effects[¥] | Q | 0.0001 | 0.7793 | 0.4443 | 0.0498 | 0.1080 | 0.4809 | 0.0106 | 0.0772 | 0.0391 | 0.0114 |
| | C | 0.0001 | 0.0414 | 0.9451 | 0.0050 | 0.9735 | 0.1748 | 0.7477 | 0.9532 | 0.1203 | 0.8539 |

**Notes.**
[1] °C, Celsius degree; WSC, water soluble carbohydrate value (Brix degree 0–25°); L*, Lightness; a*, Redness; b*, Yellowness; ΔE*, The total color difference; h, hue angle, C*, Chroma or saturation

A, B, C, D- a,b,c,d Means with the different letter within the same column are significantly different according to the Tukey's HSD test at.

Significant differences marked within columns with different superscript capital letters indicate $p \leq 0.01$.

Lowercase letters indicate $p \leq 0.05$.

[¥] L: linear; Q: quadratic; C: cubic effects.

Filik (2025), *PeerJ*, DOI 10.7717/peerj.19267

**Table 2   Nutrient content of GA silages with different additives.**

| Parameters | DM ± SEM[1–4] | OM ± SEM[2] | CA ± SEM[2] | CP ± SEM[2] | EE ± SEM[2] | CF ± SEM[2] | ADF ± SEM[2] | ADFom ± SEM[3] | NDF ± SEM[2] | NDFom ± SEM[3] |
|---|---|---|---|---|---|---|---|---|---|---|
| Control (*Guizotia abbysinica* (L. f.) Cass. (GA)) | 201.09 ± 0.01D | 96.58 ± 0.05B | 3.43 ± 0.05A | 4.57 ± 0.03AB | 8.23 ± 0.34AB | 27.88 ± 4.24AB | 40.91 ± 2.61A | 37.49 ± 2.67A | 57.91 ± 3.05A | 54.48 ± 3.10A |
| GA with 1% salt | 217.20 ± 0.18CD | 96.49 ± 0.31B | 3.52 ± 0.32A | 4.13 ± 0.01BC | 6.28 ± 0.43BC | 23.28 ± 1.70B | 37.56 ± 3.62A | 34.04 ± 3.93A | 50.23 ± 1.32AB | 46.71 ± 1.64AB |
| GA with 2.5% molasses | 248.93 ± 2.72AB | 96.46 ± 0.02B | 3.54 ± 0.02A | 4.11 ± 0.19BC | 5.34 ± 0.62C | 37.02 ± 1.17A | 25.33 ± 0.94B | 21.79 ± 0.92B | 37.65 ± 1.40BCD | 34.11 ± 1.42BCD |
| GA with 2.5% urea | 231.43 ± 1.73BCD | 97.07 ± 0.06B | 2.93 ± 0.06A | 4.02 ± 0.10BC | 6.77 ± 0.25ABC | 19.52 ± 1.42B | 15.92 ± 1.23BC | 12.99 ± 1.29BC | 27.01 ± 0.27D | 24.08 ± 0.21D |
| GA with 5% barley | 232.14 ± 13.67BCD | 96.52 ± 0.03B | 3.48 ± 0.03A | 4.77 ± 0.05AB | 6.63 ± 0.66ABC | 22.44 ± 0.01B | 18.57 ± 0.21BC | 15.09 ± 0.24BC | 34.44 ± 0.89CD | 30.96 ± 0.86CD |
| GA with 2.5% molasses + 1% salt | 268.01 ± 3.44A | 96.64 ± 0.04B | 3.36 ± 0.04A | 5.05 ± 0.02A | 5.53 ± 0.75C | 25.76 ± 1.31B | 36.58 ± 1.33A | 33.22 ± 1.29A | 43.47 ± 5.15BC | 40.11 ± 5.19BC |
| GA with 2.5% urea + 1% salt | 254.02 ± 2.43AB | 97.79 ± 0.01A | 2.22 ± 0.01B | 3.52 ± 0.01C | 9.20 ± 0.19A | 36.54 ± 1.39A | 11.05 ± 0.15C | 8.83 ± 0.16C | 31.10 ± 0.0.27CD | 28.88 ± 0.27CD |
| GA with 5% barley + 1% salt | 242.55 ± 6.23ABC | 96.63 ± 0.05B | 3.37 ± 0.05A | 5.22 ± 0.34A | 7.01 ± 0.26ABC | 21.27 ± 0.66B | 22.46 ± 0.92B | 19.09 ± 0.975BC | 36.62 ± 1.90CD | 33.25 ± 1.85CD |
| *P* values | 0.0006 | 0.0005 | 0.0005 | 0.0004 | 0.0058 | 0.0009 | 0.0001 | 0.0001 | 0.0002 | 0.0003 |
| Effects[¥]   L | 0.0012 | 0.0244 | 0.0244 | 0.0337 | 0.0378 | 0.2151 | 0.0001 | 0.0001 | 0.0001 | 0.0001 |
| Q | 0.0175 | 0.0179 | 0.0179 | 0.2571 | 0.0079 | 0.0090 | 0.1250 | 0.1885 | 0.5468 | 0.6493 |
| C | 0.0329 | 0.3113 | 0.3113 | 0.4599 | 0.5414 | 0.0004 | 0.1774 | 0.1795 | 0.5328 | 0.5070 |

**Notes.**

[1] g/kg natural material.

[2] (%) of dry matter.

[3] ADFom, ADF ash; NDFom, NDF ash.

[4] DM, In Air Dry Matter (g/kg); OM, Organic Matter (%); CA, Crude Ash (%); CP, Crude Protein (%), EE, Ether Extract (%); CF, Crude Fiber (%); ADF, Acid Detergent Fiber (%); NDF, Neutral Detergent Fiber (%)

A, B, C, D- a,b,c,d Means with the different letter within the same column are significantly different according to the Tukey's HSD test at.

Significant differences marked within columns with different superscript capital letters indicate $p \leq 0.01$.

Lowercase letters indicate $p \leq 0.05$.

[¥] L: linear; Q: quadratic; C: cubic effects.

2.5% urea + 1% salt (254.02 g/kg) compared to the control (201.09 g/kg) ($p = 0.0006$). Organic matter (OM) was also higher in the 2.5% urea + 1% salt group (97.79%) compared to other treatments ($p = 0.0005$). CP was significantly reduced in silages containing 2.5% urea + 1% salt (3.52%) compared to the control (4.57%) ($p = 0.0004$). However, ether extract (EE) was highest in the 2.5% urea + 1% salt group (9.20%), suggesting an improvement in EE content ($P = 0.0058$). Acid detergent fiber (ADF) and neutral detergent fiber (NDF) values significantly decreased with the addition of urea and molasses, indicating better fiber digestibility ($p < 0.0001$).

### Energy values of GA silages

The addition of different additives significantly influenced the energy values of GA silages (Table 3). The metabolizable energy (ME) was highest in the 5% barley + 1% salt (1.97 Mcal/kg) and 2.5% molasses + 1% salt (1.95 Mcal/kg) groups compared to the control (1.92 Mcal/kg) ($p = 0.0003$). Similarly, the net energy for lactation (NEL) was highest in the 5% barley + 1% salt group (1.21 Mcal/kg), followed by 2.5% molasses + 1% salt (1.20 Mcal/kg), demonstrating improved energy availability ($p = 0.0001$).

### Relative feed value and forage quality of GA silages

RFV and RFQ were significantly improved with the addition of urea and barley (Table 4). The highest RFV was observed in silages containing 2.5% urea (263.47) and 2.5% urea + 1% salt (240.21), compared to the control (92.04) ($p = 0.0001$). Likewise, RFQ was highest in the 2.5% urea group (192.28), indicating superior forage quality ($p = 0.0041$).

## DISCUSSION

There have been many studies evaluating GA cake, obtained after the extraction of its rich oil, in terms of animal nutrition. *Chavan (1961)* reported that silage from the GA plant is not suitable for cattle feeding but can be used for sheep. Apart from the study by *Peiretti, Gai & Tassone (2015)*, *Szuba-Trznadel et al. (2022)* and *Szuba-Trznadel et al. (2024)*, which evaluated the GA plant as roughage and silage in different vegetation periods in terms of animal nutrition, there is no reference with different additives to its silage potential. This study will be the first reference investigating the silage potential of the GA plant harvested during the early flowering period with the addition of urea, salt, molasses, or barley additives.

The quality of fermentation in silage is determined by calculating the Flieg score, with pH being the most influential parameter. A high dry matter (DM) content in the silage material can negatively affect silage quality by reducing acidity and inhibiting the formation of LABc (*Gürsoy, Sezmiş & Kaya, 2023*). The optimal pH range for high-quality silage is between 3.70 and 4.20 (*Kung & Shaver, 2001*). In addition to pH, DM content, WSC content, and the Flieg score are important parameters for silage quality. In this study, the differences among treatments in these characteristics were found to be statistically significant. The pH of the treated silages decreased compared to the control, and this reduction positively influenced silage quality. As reflected in the Flieg scores, the treatments improved silage quality by enhancing pH, DM content, WSC content, and overall fermentation. The best

Filik (2025), *PeerJ*, DOI 10.7717/peerj.19267

**Table 3  Energy values of GA silages with different additives.**

| Parameters[1,2,3] | NFE ± SEM | NFC ± SEM[1] | TC ± SEM[1] | DE ± SEM | ME ± SEM | NE$_L$ ± SEM | NE$_M$ ± SEM | NE$_G$ ± SEM |
|---|---|---|---|---|---|---|---|---|
| Control (*Guizotia abbysinica* (L. f.) Cass. (GA)) | 51.55 ± 3.80BCD | 25.87 ± 2.25D | 83.77 ± 0.36B | 2.35 ± 0.01ABC | 1.92 ± 0.01BC | 1.19 ± 0.01B | 1.08 ± 0.01AB | 0.53 ± 0.01B |
| GA with 1% salt | 58.66 ± 1.64AB | 35.86 ± 3.71CD | 86.08 ± 0.09AB | 2.34 ± 0.00BC | 1.92 ± 0.00BC | 1.18 ± 0.00B | 1.08 ± 0.01AB | 0.52 ± 0.00BC |
| GA with 2.5% molasses | 45.98 ± 0.37CD | 49.37 ± 0.58ABC | 87.02 ± 0.83A | 2.30 ± 0.02CD | 1.88 ± 0.01CD | 1.16 ± 0.01C | 1.04 ± 0.01BC | 0.49 ± 0.01CD |
| GA with 2.5% urea | 63.26 ± 1.87A | 59.27 ± 0.02A | 86.28 ± 0.29AB | 2.35 ± 0.00AB | 1.92 ± 0.00BC | 1.18 ± 0.00B | 1.08 ± 0.00AB | 0.53 ± 0.01B |
| GA with 5% barley | 59.17 ± 0.55AB | 42.60 ± 1.54BC | 85.11 ± 0.64AB | 2.37 ± 0.00AB | 1.95 ± 0.01AB | 1.20 ± 0.00AB | 1.10 ± 0.00A | 0.55 ± 0.01AB |
| GA with 2.5% molasses + 1% salt | 55.45 ± 0.83ABC | 49.49 ± 0.60BC | 86.06 ± 0.73AB | 2.38 ± 0.00AB | 1.95 ± 0.01AB | 1.20 ± 0.00AB | 1.11 ± 0.01A | 0.55 ± 0.01AB |
| GA with 2.5% urea + 1% salt | 44.77 ± 1.54D | 53.97 ± 0.07AB | 85.06 ± 0.20AB | 2.28 ± 0.00D | 1.87 ± 0.01D | 1.14 ± 0.00C | 1.02 ± 0.00C | 0.47 ± 0.00D |
| GA with 5% barley + 1% salt | 59.38 ± 0.30AB | 47.79 ± 1.35ABC | 84.41 ± 0.55AB | 2.40 ± 0.02A | 1.97 ± 0.02A | 1.21 ± 0.01A | 1.12 ± 0.01A | 0.56 ± 0.01A |
| | *P* values | 0.0005 | 0.0002 | 0.0307 | 0.0003 | 0.0003 | 0.0001 | 0.0003 | 0.0001 |
| Effects[¥] | L | 0.0203 | 0.0001 | 0.0068 | 0.5010 | 0.2907 | 0.0727 | 0.3395 | 0.3052 |
| | Q | 0.0192 | 0.9866 | 0.0198 | 0.0136 | 0.0353 | 0.0085 | 0.0193 | 0.0114 |
| | C | 0.0002 | 0.5342 | 0.8980 | 0.0110 | 0.0094 | 0.0068 | 0.0159 | 0.0111 |

**Notes.**

[1] (%) of dry matter.

[2] Data represent the mean of four applications of each treatment.

[3] NFE, nitrogen-free extract (g/kg); NFC, non-fibre carbohydrates (g/kg); TC, total carbohydrates (g/kg); DE, digestible energy (Mcal/kg); ME, Metabolizable energy (ME Mcal/kg); NE$_L$, net energy-lactation (Mcal/kg); NE$_M$, net energy-maintenance (Mcal/kg); NE$_G$, net energy-gain (Mcal/kg)

A, B, C, D- a,b,c,d Means with the different letter within the same column are significantly different according to the Tukey's HSD test at.

Significant differences marked within columns with different superscript capital letters indicate $p \leq 0.01$.

Lowercase letters indicate $p \leq 0.05$.

[¥] L: linear; Q: quadratic; C: cubic effects.

**Table 4 Relative feed value and relative forage quality of silages from GA with different additives.**

| Parameters[1,2,3] | DDM ± SEM | DMI ± SEM | TDN ± SEM | RFV ± SEM | RFQ ± SEM |
|---|---|---|---|---|---|
| Control (*Guizotia abbysinica* (L. f.) Cass. (GA)) | 57.03 ± 2.047C | 2.08 ± 0.11E | 53.21 ± 0.27ABC | 92.04 ± 8.10E | 89.92 ± 5.17E |
| GA with 1% salt | 59.65 ± 2.82C | 2.39 ± 0.06DE | 53.08 ± 0.11BC | 110.69 ± 8.14E | 103.17 ± 2.52DE |
| GA with 2.5% molasses | 69.17 ± 0.73B | 3.19 ± 0.12BCD | 52.10 ± 0.27DC | 171.07 ± 4.58CD | 135.21 ± 5.77BCD |
| GA with 2.5% urea | 76.50 ± 0.96AB | 4.45 ± 0.04A | 53.23 ± 0.00ABC | 263.47 ± 0.66A | 192.28 ± 1.94A |
| GA with 5% barley | 74.44 ± A0.16AB | 3.49 ± 0.09BC | 53.81 ± 0.05AB | 201.19 ± 4.79BC | 152.52 ± 4.10BC |
| GA with 2.5% molasses + 1% salt | 60.41 ± 1.04C | 2.80 ± 0.33CDE | 53.86 ± 0.11AB | 130.88 ± 13.30DE | 122.60 ± 14.29CDE |
| GA with 2.5% urea + 1% salt | 80.29 ± 0.12A | 3.86 ± 0.03AB | 51.51 ± 0.09D | 240.21 ± 2.40AB | 161.62 ± 1.09AB |
| GA with 5% barley + 1% salt | 71.41 ± 0.72B | 3.29 ± 0.17BC | 54.35 ± 0.41A | 181.78 ± 7.61BC | 145.23 ± 8.61BC |
| *P* values | 0.0001 | 0.0001 | 0.0002 | 0.0001 | 0.0041 |
| Effects[¥]   L | 0.0001 | 0.0001 | 0.3503 | 0.0001 | 0.0001 |
| Q | 0.1251 | 0.0134 | 0.0161 | 0.0009 | 0.0119 |
| C | 0.1783 | 0.9595 | 0.0128 | 0.7712 | 0.8427 |

**Notes.**
[1] (%) of dry matter.
[2] Data represent the mean of four applications of each treatment.
[3] DDM, digestible dry matter (%); DMI, dry matter intake (live weight: LW, %); TDN, total digestible nutrients (%); RFV, relative feed value; RFQ, relative forage quality
A, B, C, D- a,b,c,d Means with the different letter within the same column are significantly different according to the Tukey's HSD test at.
Significant differences marked within columns with different superscript capital letters indicate $p \leq 0.01$.
Lowercase letters indicate $p \leq 0.05$.
[¥] L: linear; Q: quadratic; C: cubic effects.

silage treatments were those containing 2.5% urea + 1% salt and 2.5% molasses + 1% salt. The addition of 1% salt altered the osmotic pressure in GA silage, leading to an increase in WSC content. This increase in WSC content was associated with a rise in silage pH, ultimately achieving the desired pH range for high-quality silage. Furthermore, urea and molasses provided essential energy sources for lactic acid bacteria to sustain their metabolic activities. Among these two treatments, the best LABc growth was observed in the silage supplemented with urea. In this treatment, WSC supplied the necessary energy for bacterial activity, while urea provided nitrogen. Conversely, in the molasses-treated silage, the high WSC content and increased energy availability from molasses created a favorable environment for LABc and other beneficial bacteria.

As expected, the highest WSC content was found in the 2.5% urea + 1% salt silage (18.50). The WSC contents of 2.5% molasses + 1% salt (18.25), 2.5% urea (16.75), and 5% barley (16.00) were not significantly different from the 2.5% urea + 1% salt treatment, indicating their similar effects on silage fermentation and quality. Silages with Fleig scores between 100 and 80 are classified as excellent, those scoring between 80 and 61 are considered good, and silages falling within the 60 to 41 range are categorized as average in quality. A score between 40 and 21 indicates low-quality silage, while scores from 20 to 0 denote very poor quality (*Kılıç, 1986*). According to Fleig scores, the silages containing 2.5% molasses + 1% salt (93.60), 2.5% urea + 1% salt (93.61), 2.5% molasses (86.41), 5% barley + 1% salt (86.41), and 2.5% urea (82.79) were classified as excellent in terms of silage quality. In contrast, the 5% barley (75.53), 1% salt (73.24), and control treatment (69.82) were categorized as good ($p < 0.0001$). The results indicate that all additives positively

influenced silage quality when compared to the control treatment ($p = 0.0001$). The color values measured four times from the silage surface are expressed as L*: lightness (black: 0 - white: 100), a*: red to green (+a*: red, -a*: green) and b*: yellow to blue (+b*: yellow, -b*: blue). When the L*; a*; and b* values of the silage were examined in the color values, it was found that the silage had a dark green color (Table 1), which was determined in the study as 25.36, 25.38, 24.48, 25.56, 24.82, 22.70, 24.38, and 25.38 ($p < 0.01$); 3.07, 2.90, 3.39, 2.84, 2.86, 2.74, 3.25 and 2.65 ($p > 0.05$); and 8.34, 7.30, 6.99, 7.97, 6.78, 6.63, 7.68 and 7.48 ($p < 0.05$), respectively. While *Çayıroğlu et al. (2020)* reported a medium dark color with increased yellowing in their study on sugar beet silage, *Filik & Filik (2021a)* found that Amaranths silages, produced with similar additives, had a dark green color, consistent with the findings of the present study.

In the present study, the addition of molasses further increased the energy demand of the bacteria in the silage and enabled them to utilize EE, which is another readily available energy source, to meet their energy requirements. In treatments where urea was added (2.5% urea and 2.5% urea + 1% salt), the required energy was primarily provided by ADF and NDF, which are structural carbohydrates, due to the lack of readily available energy sources. In contrast, in the 2.5% urea + 1% salt treatment, it was more evident that salt facilitated energy utilization by disrupting the cell structure. The NFC (non-fiber carbohydrates) ($p < 0.01$) and TC (total carbohydrates) ($p < 0.05$) values in the control treatment were positively influenced by the treatments, with the 5% barley + 1% salt treatment showing the highest DE, ME, NEL, NEM, and NEG values. This improvement can be attributed to the addition of salt, which disrupted the cell structure of barley—a carbohydrate-rich and easily degradable feed ingredient—resulting in an increased release of carbohydrates and higher silage energy values. *Szuba-Trznadel et al. (2022)* conducted a two-year study to determine the optimal harvest time for GA forage. This was achieved by analyzing the nutrient composition of post-harvest forage on the 90th day, corresponding to the plant's early flowering period. These findings align with those of the present study. However, an increase in cellulose fractions was observed in the ensiled samples of the current study, particularly when compared to the control group, due to climatic factors. As demonstrated in previous forage studies by *Szuba-Trznadel et al. (2024)*, climate has a similar impact on silage quality. When comparing GA harvested after 90 days in Poland with the current study, cellulose fractions remain lower in the Polish samples. The primary factor contributing to this discrepancy is believed to be the variation in daytime temperature, humidity, and rainfall during the summer months, despite similar average temperature data between the two locations. As a result, the cellulose cell wall of GA cultivated in Türkiye appears to have developed a structural form influenced by temperature. Considering these findings, the additives incorporated in this study have been shown to positively impact silage quality, supporting the working hypothesis.

The microbial growth is shown in Fig. 1. The 2.5% urea + 1% salt treatment had the lowest energy value, while LABc showed the highest growth in this treatment. The increase in LABc was attributed to the addition of urea and salt to the silage. Urea provided the necessary nitrogen (N) source for LABc growth, while salt facilitated light energy availability by disrupting the cell structure and lowering the pH, thereby preventing yeast

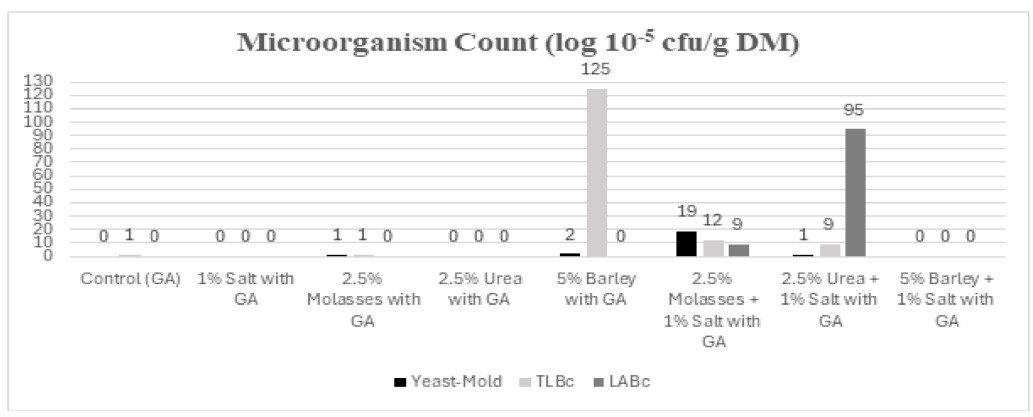

Figure 1 **Microbial counts of GA silage treatments.** LABc, Lactic acid bacteria count and TLBc, Total live bacteria count.

and mold formation (*Yunus et al., 2000*). *Canbolat, Kamalak & Kara (2014)* reported that urea supplementation in pomegranate pulp silage prevented yeast and mold formation, reduced pH, and enhanced LABc development, which aligns with the findings of the present study. As this is the first publication on GA silage, RFV and RFQ were evaluated and presented in Table 4 to determine the extent to which plant silage can meet animal nutritional needs under *in vitro* conditions. Specifically, RFV was calculated based on ADF and NDF values to assess energy content and roughage quality (*Filik & Filik, 2021a*; *Filik & Filik, 2021b*).

According to the RFV values, the control treatment was classified as Quality III, while the 1% salt and 2.5% molasses + 1% salt treatments were categorized as Quality II. The highest-quality silages were obtained from the 2.5% molasses, 5% barley + 1% salt, 5% barley, 2.5% urea + 1% salt, and 2.5% urea treatments ($p = 0.0001$). The study concluded that silage quality improved in treatments that provided essential energy and protein sources for sustained bacterial fermentation, with the highest RFV value observed in the 2.5% urea silage.

Another important parameter is RFQ, which was developed to assess feed suitability based on animal performance. RFQ is calculated using DMI (dry matter intake), which estimates the amount of dry matter consumed by the animal, and TDN, which determines the total amount of digestible nutrients available from the feed (*Filik & Filik, 2021a*; *Filik & Filik, 2021b*). Based on the RFQ values, silages from 2.5% urea and 2.5% urea + 1% salt appeared to be of the highest quality but are recommended only for dry cows aged 18 to 24 months. In contrast, silages from 5% barley, 2.5% molasses + 1% salt, and 5% barley + 1% salt were deemed suitable for dairy cows and first-trimester dairy calves. Silages from the control, 1% salt, and 2.5% molasses treatments were found to be appropriate for dairy cows, heifers in the last 200 days of gestation, and stocker cattle aged 3 to 12 months. The RFQ values determined in the study were 89.92, 103.17, 135.21, 192.28, 152.52, 122.60, 161.62, and 145.23, respectively ($p = 0.0041$).

This study is the only reference examining the usability of the GA plant harvested during the early flowering period with the addition of urea, salt, molasses, or barley to evaluate silage quality. In determining silage quality, the Flieg score is calculated, with pH identified as the most influential parameter. A high DM content in silage material can negatively impact silage quality by reducing acidity and inhibiting LABc formation. The optimal pH range for high-quality silage is between 3.70 and 4.20. In addition to pH, factors such as DM content, WSC content, and the Flieg score are critical for evaluating silage quality. Statistically significant differences were observed among the treatments in these parameters. The addition of urea and molasses stimulated LABc growth and provided the necessary energy to sustain bacterial activity in silage. Among these treatments, the highest LABc growth was observed in the urea-added treatment. In the molasses-added silage, the high WSC content and increased energy availability created an environment conducive to the growth of LABc and other beneficial bacteria. In the urea + salt treatment, the silage pH increased, allowing the desired pH range to be achieved. Higher energy requirements facilitated bacterial utilization of the silage. In urea-treated silages, the required energy was primarily provided by structural carbohydrates, such as ADF and NDF. The significant LABc development observed in urea-treated silages was attributed to urea providing an essential nitrogen source for bacterial growth and salt inhibiting yeast and mold formation by disrupting the cell structure and lowering the pH. Additionally, the addition of salt and molasses to GA silage increased its overall energy value.

## CONCLUSION

This study demonstrated that ensiling *Guizotia abyssinica* (L. f.) Cass. with various additives significantly improved fermentation quality, nutrient composition, and energy values. The highest Fleig scores were observed in 2.5% urea + 1% salt (93.61) and 2.5% molasses + 1% salt (93.60), classifying them as excellent-quality silages. The pH values of these treatments were 4.06 and 4.13, respectively, falling within the optimal silage pH range of 3.70–4.20. Additionally, the highest WSC content was recorded in 2.5% urea + 1% salt (18.50), followed by 2.5% molasses + 1% salt (18.25), indicating enhanced fermentability. Nutritional assessments revealed that GA silage supplemented with 2.5% urea exhibited the highest RFV (263.47) and RFQ (192.28) values, making it most suitable for dry cows aged 18–24 months. In contrast, 5% barley + 1% salt silage, with a metabolizable energy (ME) of 1.97 Mcal/kg, was better suited for dairy cows and first-trimester dairy calves. The study also found that GA forage ensiled with different additives significantly enhanced crude protein (CP) content, ranging from 3.52% to 4.57%, while reducing acid detergent fiber (ADF) and neutral detergent fiber (NDF) levels, thereby improving digestibility. These findings suggest that the addition of molasses, urea, barley, and salt significantly enhances GA silage quality, with 2.5% urea emerging as the most effective additive. While urea-treated silage demonstrated the highest nutritional value, its use should be limited to dry cows aged 18 to 24 months, whereas other treatments may be more suitable for different animal species at various growth stages. The study concluded that GA silage, when prepared with common additives such as barley, salt, molasses, or urea, can serve as a viable

silage material under *in vitro* conditions. However, further research involving large-scale silage production and alternative additive formulations could increase its practicality and appeal to farmers. Given the growing global emphasis on sustainable agriculture, GA presents significant potential as an alternative silage crop, particularly in regions affected by water scarcity. Future studies should focus on *in vivo* digestibility trials and large-scale assessments to fully establish the practical applications of GA silage in ruminant nutrition.

## ACKNOWLEDGEMENTS

The author would like to thank the Faculty of Agriculture and the students of the Department of Agricultural Biotechnology.

### Funding
This work was supported by Kırşehir Ahi Evran University Scientific Research Commission (Grant numbers ZRT.A4.19.007). The funders had no role in study design, data collection and analysis, decision to publish, or preparation of the manuscript.

### Grant Disclosures
The following grant information was disclosed by the author:
Kırşehir Ahi Evran University Scientific Research Commission: ZRT.A4.19.007.

### Competing Interests
The author declare that she has no competing interests.

### Author Contributions
- Ayşe Gül Filik conceived and designed the experiments, performed the experiments, analyzed the data, prepared figures and/or tables, authored or reviewed drafts of the article, and approved the final draft.

### Data Availability
   The raw data is available in the Supplemental File.

### Supplemental Information
Supplemental information for this article can be found online at http://dx.doi.org/10.7717/peerj.19267#supplemental-information.

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
