# Peer review of "Nutritional value of ensiled Guizotia abyssinica (L. f.) Cass. (Noug: Niger) treated with salt, molasses, urea or barley"

_PeerJ, doi:10.7717/peerj.19267_

## Round 0.1 · original submission · Major Revisions

After three considerable reviews, the consensus appears to be a decision for major revision. However, this was close to a rejection for me. In light of this, I would urge the authors to carefully and thoroughly consider the reviewers' remarks and address each as comprehensively as possible, not merely make editorial changes. The introduction, context, and rationale for the study need to be set up more thoroughly, and so do the discussion and conclusion.

Reviewer 1 ·

Basic reporting

1. English language can be improved, maybe during revision process Author can send manuscript to correction by the native speaker.

2. More literature references must be added in the manuscript, especially from last 5 years.

3. References section must be checked by Author, because three references in lines 331, 381 and 434 are not mentioned in the main text of manuscript. From the other side references emphasized in lines 85, 117, 139, 184 are not added to the References section.

4. Structure of manuscript must be improved according to mean article structure in PeerJ. Lack of Conclusions section. Results and Discussion are merged by Author in one section ‘Results and Discussions’, it is better to divide them, because at this moment this merge provide a lot of confusion during reading of the text. Subsections in case of these two sections will also can solve the problem with readability of text.

5. Figure 1 must be improved. Linear relation in this case is not logical, better is to use column chart, additionally unit which is present in title log 10-5 cfu/g of microorganisms is very low.

6. Statistical analysis description/subsection must be improved. In tables significance of differences using post-hoc test are described after one-way ANOVA test between 8 treatments. SED value is problematic in this case, maybe better is to use SEM.

7. Raw Data. DOE is correct prepared to input it in one-way ANOVA design, but 8 treatments and 4 replications gives power of a test about 0.11 (where 1.0 is maximal), that could increase risk of error during statistical analysis.

8. Manuscript is self-contained because Author described entire process from sowing of Guizotia abyssinica seeds, preparing raw material to ensiling process and nutritive value of obtained silages in 8 different variants, conducted microbial analysis and determined its relative feed value and relative feed quality from point of view in vito experiment. In consequence next step necessity of experiment with animals appears.

Experimental design

1. Manuscript is appropriate to evaluate during review and revision process in PeerJ.

2. Author submitted Research Article, for which DOE was planned before its realization (power of a test is not high, but even Fisher started from lower number of replications in treatments).

3. Article besides of older references is novel because as Author described, there are not many publications from this subject, first attempts of ensiling of Guizzotia abbysinica was made, but for sure addition of additional source of nitrogen in silage will increase its nutritive value with very low cost.

4. The research questions are defined well, in manuscript is lack higher number references from last 5 years, but solution proposed by Author fulfilling the knowledge gap in case of manner how to improve valuable silage for ruminants (of course amount of urea in diet for ruminants will be also important in in vivo experiment). In this case question is how many silage can be offered to animals, what can be limited factor for them: intake of the silage (DM) or substances present in silage, which can become antinutritive with increasing intake of this feed.

5. Materials and Methods section is described quite well, but of course subsections will be useful in this place to increase readability of text. Statistical analysis fragment/subsection must be improved. Author used one-way ANOVA test, but it wasn’t mention in description of it. Question appears also when post-hoc test is taking into a consideration. Author used Duncan’s test, but when 8 treatments was planned, maybe better is to use more conservative HSD Tukey’s test.

Validity of the findings

Results proposed in experiment additives are very perspective, but:

1. More references are required from last five years.

2. Statistical analysis must be described more precise.

3. Explanation of 4 replications in treatment is required.

4. Conclusions must be added by Author.

5. In my opinion separate sections must be stated.

6. Major revision is required in my opinion.

Additional comments

Nutritional value of ensiled Guizotia abyssinica (Nough: Niger) treated with salt, molasses, urea or barley

Dear Author,
text of manuscript is interesting because shows how to increase nutritive value of silage from Guizzotia abyssynica, it contents a high amount of carbohydrates, but nitrogen is limiting factor considering its value for example cattle, and in this case additional amount of urea in it can affect positively for fermentation profile in ruminants. Of course after in vitro analysis of quality of silages from different treatments, in vivo experiment with animals will be important to use this feed in ruminant nutrition, and as was described even in case monogastric animals.

Below I added some suggestions helpful in revision process:

Line 27
In text of manuscript is given experimental groups, but maybe treatments will be more precise word in this case, and in entire manuscript?

Line 36
Italics can be added to: in vitro and in vivo. Please check in entire text.

Line 37
Maybe instead of Niger, better will be binomial nomenclature name Guizotia abbysinica, because it have larger spectrum during searching of scientific articles.

Line 54
Alemaw and Wold, according to References section year of publication should be: 1991. The same situation with Alemayehu and Ashagrie, the same Workshop Book Pages (1991).

Line 62
Alemaw and Wold, according to References section year of publication should be: 1991.

Line 66
Without 2004, and without closing the brackets after Galmessa et al. 2004, with following after references?

Lines 72-75
There are also several new publications from last 5 years in which despite of Peiretti et al. 2015, Guizzotia abbysinica was introduced in European conditions, with first experiments in which nutritional value of this plant was also determined in different days after sowing, but without any additives during the preparation of silage.

Line 73
Getinet and Sharma, 1996.

Lines 79- 135
Maybe it is possible to divide Materials and Methods on subsections describing for example condition and gathering of samples, preparation of silages, chemical analyses, statistical analysis, …

Line 85
Lack of the one most important scientific article! (Peiretti et al. 2015) in References section.

Lines 88-90
In experiment 8 treatments was used, each in four replications. That can gives low power of a test because for those 32 replications in this kind of DOE. In this case 1-ß is equal about 0.11, maybe in next experiment possible will be increase number of replication in groups or reduction of treatments, what will increase number of replications.

Line 91
In text of manuscript are emphasized treatment groups, but in this case simple treatments will be appropriate.

Line 114
AOAC procedures can be also emphasized in References section.

Line 115
In text of manuscript Crude cellulose (CF), but in this case perhaps Crude fibre (CF) must be used.

Line 117
Van Soest et al. 1991, reference must be added in References section.
References for Ankom procedures can be also added in References section (2016, 2017 a-c).

Lines 131-135
Statistical analysis subsection must be rewrote, because there is information about descriptive statistics using “Descriptive variables” (I heard about independent and dependent variables, but in case of descriptive not).
Also in case of tables SEM (standard error of mean) value was used, but in analysis SED is described. In this case better to use mentioned SEM or SE.
In tables was used one-way ANOVA test, that is why in my opinion better is to add information about it in text, after it Duncan’s test can be conducted (but there are also 8 treatments, and maybe HSD Tukey’s test will be more appropriate in for this DOE).
In this place also information about normality distribution in treatments (Shapiro-Wilk’s test) and homogeneity of variance (Levene’s test) between treatments are required.
Calculations are conducted in case of different dimensions, but maybe is also possible to present differences in case linear, quadratic or cubic effect? In tables are only emphasized differences using post-hoc test after one-way ANOVA test.

Line 137
In my opinion better will be to separate ‘Results and Discussions’ section to Results and Discussion.
Subsections for each can be also added to divide entire text (the same as in published scientific articles on PeerJ site).
During comparing different mean values between treatments, can be also added in brackets.

Line 139
Reference for Chavan 1961 must be added.

Lines 140-141
Checking Reference section, there are not many publications from last 3years, where quality of roughage or quality of ensiled Guizzotia abbysinica could be also determined.

Lines 147, 151-152
Significance level required.

Lines 151-152
The same as in lines 144-147.

Line 154-155
English form also can be checked in manuscript.

Lines 154-178
Significance level and mean values from compared treatments can be also added in text.

Lines 165-167
Significance level required.

Lines 171, 182, 193 and 210
In statistical significance level p-value must be used. Sample from population was randomly choose.

Lines 173-178
These values are mainly emphasized during description of meat quality, but maybe it is possible to determine correlation between its and amount of intake silage.

Line 184
Please add in References section scientific article prepared by Dumlu Gül and Tan, 2021.

Lines 185-190
Significance level and mean values from compared treatments can be also added in text.

Lines 198-200
Significance level description will be useful information there, but my opinion column chart will be better option there, because there is no logical relation with 2.5% urea with Niger and 5% of Barley (to TLBc), when additive is completely different, that could have sense when amount of urea increase to 5%.

Line 202
Canbolat et al. (2014) .

Lines 219-224
Maybe it is possible to add reference(s) in this part of manuscript, p-value in case of comparison can be also added.

Line 251
Conclusions section required.

Line 264
References section
More references must be added from last 3-4 years.
References must be adapted to PeerJ pattern (Instructions or comparison with published scientific articles in PeerJ).

Line 331
Lack of reference Dempewolf et al. (2010) in main text of manuscript.

Line 381
Lack of reference Jeranyama and Garcia (2004), perhaps can be added in lines 124-125.

Line 434
Lack of reference Steel and Torrie (1980), perhaps can be added in lines 131-135.

Figure 1
In my opinion column chart will be more appropriate to present Microbial counts, because there is no logical relation with 2.5% urea with Niger and 5% of Barley (to TLBc), when additive is completely different, that could have sense when amount of urea increase to 5%.
Please also check if microorganism count log 10-5 cfu/g is real (there should be a lot of microorganisms, log 105 cfu/g )?

Tables

Line 1
Table 1
In fifth row from the bottom is SED described as standard error of the difference between two means, but in this case SED must be calculated between each pair of mean values for all treatments. In this case better will be to use SEM (standard error of mean), which describe relation between all treatments.
Maybe is possible to describe differences between effects between treatments when model is significant, but there is also left question, which effect is the best to description dependent variable when more than one effect is statistically significant.

Line 3
In line 4 significance level for p<0.01 is described, in this case Capital letters must be used (A,B,C,D), or in description in line 4 significance level can be also changed for p<0.05 and normal letters can be used as described.

Line 5
SEM instead of SED must be used.

Line 34
Header of Table 2, Ash is mentioned, but specific ash or Crude ash?
The same as in Line 1 (Tables).

Line 36
Ash or (CA) Crude ash?

Line 37
In case of letters in superscript e must be added (a,b,c,d,e) or (A,B,C,D,E) depends of significance level, respectively p<0.05 or p<0.01.

Line 38
The same as in line 3.

Line 39
SEM instead of SED must be used.

Line 67
Table 3
The same as in Line 1 (Tables).

Line 71
In case of letters in superscript e must be added (a,b,c,d,e) or (A,B,C,D,E) depends of significance level, respectively p<0.05 or p<0.01.

Line 72
SEM instead of SED must be used.

Line 100
Table 4
The same as in Line 1 (Tables).

Line 103
In case of letters in superscript e must be added (a,b,c,d,e) or (A,B,C,D,E) depends of significance level, respectively p<0.05 or p<0.01.

Line 104
SEM instead of SED must be used.

·

Basic reporting

The study highlights the significance of the niger crop in silaging, a topic of considerable relevance in the current context. While the authors have conducted a commendable study, the presentation of the manuscript falls short of expectations. A thorough revision is necessary to enhance its quality. These are my observations after reviewing the manuscript.

1. The manuscript contains several grammatical errors, and the overall language quality requires significant improvement for better readability and clarity.
2. In the introduction, the authors should include information on the status of niger cultivation in Turkey, as the study is conducted there. Additionally, providing statistical data on the global production and area occupied by this crop would add valuable context to the study.
4. The author mentions in the introduction that the results of the present study were compared with the sunflower population. However, this seems irrelevant, as there is no evidence of any study conducted by the author on the sunflower population. Unless the author has directly conducted such a study, it is unnecessary to reference sunflower in the introduction.
5. Since your study focuses on the use of silage for animal feed, it is crucial to include information on the nutrient content of niger cake, as it is directly relevant. While the nutrient content of niger oil is provided, it holds less significance in the context of this study.
6. Please ensure consistency in the reference listing by following the same format for all references. Uniformity in citation style will enhance the manuscript’s professionalism and readability. Also, please crosscheck the references thoroughly.
7. References like Peiretti et al. (2015), Van Soest et al. (1991), Ankom (2016, 2017a, 2017b, 2017c), and Chavan (1961) are cited in the text but are missing from the reference section.
8. Jeranyama, P., and Garcia, A.D. (2004) are listed in the reference section but not cited in the text.

Experimental design

1. Please mention the cultivar of niger used in the study or was it a random selection? The results of such studies may vary depending on the genotype used. So, it is always better to mention the name of the cultivar and why you selected this particular genotype.

2. In line #106, the Fleig scale ranges are mentioned as 100–80 and 80–61. However, in line #168, the Fleig scores are given as 80–100 and 61–80. To avoid confusion and ensure clarity for the reader, it is important to maintain consistency in presenting these ranges throughout the manuscript.

3. It is recommended to provide the full form of each abbreviation the first time it is mentioned in the manuscript, followed by the abbreviation in parentheses. For example, abbreviations like WSC, DM, and LABc should be introduced with their full forms initially to ensure clarity for the reader.

Validity of the findings

1. Some sections, particularly the Results and Discussion, are overly cluttered, making them difficult to follow. It is advisable to divide the Results and Discussion sections using subheadings. This would enhance readability and make it easier for readers to follow the key points and findings.

2. In line #220, the manuscript mentions the results of feeding the treated silage to cattle. However, is there any data to support these claims, such as changes in the animals' weight or milk production? Including such evidence would strengthen the study's conclusions.

3. The claims made by the author regarding the results of the study are quite weak, and there is a lack of strong supporting evidence to substantiate these claims. It is recommended to strengthen these statements with more robust data or references to enhance the credibility of the results.

Additional comments

I recommend restructuring the entire manuscript, improving the language, and presenting the content in a more scientific and formal manner. This would enhance the clarity and overall quality of the manuscript.

·

Basic reporting

The manuscript requires major corrections to be upgraded.

Experimental design

It lacks clarity and information.

Validity of the findings

The findings need to be reported based on the significance and justification should be provided behind the outcome.

Additional comments

Authors can follow the specific comments in the manuscript and upgrade it accordingly.

---

## Round 0.2 · Minor Revisions

The revised manuscript has been reviewed, and a decision of minor revision has been made. Thank you for your efforts in revision this submission.

Reviewer 1 ·

Basic reporting

a) After major revision the manuscript was significantly improved, which on the one hand made it clearer and more coherent and additionally increased its cognitive value. In my opinion, minor corrections are needed from the point of view of text editing.

b) The manuscript has been supplemented with articles from the last few years, in my opinion this is absolutely sufficient, number of references was increased to 81. In the main part of the manuscript there are 81 references listed. Reference Szuba-Trznadel et al. 2024 is mentioned in Discussion section, but it is not included in the References section. Additionally, a minor correction will be needed in line 552, the second author is Shaver R. (in the manuscript it is Shave).

c) The manuscript has been significantly improved, subsections have been added to make the text more readable, and chapters can be further distinguished by the standard use of Capital letters in section title. Significant differences have been highlighted in tables, and SEM has been added to the mean values for treatments.

d) The manuscript is self-contained, it contains a classic division into parts from Introduction to Conclusions formulated on the basis of design of experiment (DOE) and conducted research. Reference section can be quickly corrected.

Experimental design

a) The current form of the manuscript fits within the publication scope of PeerJ.

b) The research problem is as appropriately defined. Additionally, it goes beyond the scope of the research of previous authors, where the conditions of sowing and growth of Guizotia abbysinica in different climatic conditions, the obtained yield of green fodder and the possibility of its ensiling were taken into account.

c) The manuscript develops earlier research directly towards improving the nutritional/nutritional value of the prepared silage based on various additives, which will certainly enable the transfer of research results to nutritional practice.

d) The DOE is correctly prepared, what increase power of a test and allowed the formulation of conclusions based on the obtained results, which will certainly be useful in practice. Statistical analysis supports the Conclusions. No reservations from an ethical point of view.

Validity of the findings

a) The research expands the scope of knowledge on the possibilities of using Guizotia abbysinica silage with additives improving its nutritional value in practical nutrition of animals.

b) The data provided are robust, statistically sound and controlled due to the preparation of the design of the experiment.

c) The conclusions are well stated and explain the purpose of the experiment described on the end paragraph of Introduction

Additional comments

Several suggestions helpful during the minor revision.

Lines 43, 104, 190, 235, 362
Section titles must be emphasized by capital letters.

Line 181
Shapiro-Wilk’s test.

Line 186
Tukey’s HSD multiple range test.

Lines 239 and 302
Lack of reference Szuba-Trznadel et al. 2024 in References section.

Line 362
Conclusions must be used.

Line 552
In text of manuscript are mentioned ‘…Kung L, Shave R. 2001…’, must be ‘…Kung L, Shaver R. 2001…’.

In description of Tables 1-3 Tukey’s HSD test also must be used. (Saxon genitive).

Figure 1. Resolution must be increased.

---

## Round 0.3 · accepted · Accept

Your revised submission has been accepted.